# Evaluating the kidney disease progression using a comprehensive patient profiling algorithm: A hybrid clustering approach

Mohammad A. Al-Mamun[1]*, Ki Jin Jeun[1], Todd Brothers[2], Ernest O. Asare[3], Khaled Shawwa[4], Imtiaz Ahmed[5]

1 Department of Pharmaceutical Systems and Policy, West Virginia University, Morgantown, West Virginia, United States of America, 2 Department of Pharmacy Practice, University of Rhode Island, Kingston, Rhode Island, United States of America, 3 School of Public Health, Yale University, New Haven, Connecticut, United States of America, 4 Department of Nephrology, West Virginia University, Morgantown, West Virginia, United States of America, 5 Department of Industrial and Management Systems Engineering, Morgantown, West Virginia, United States of America

* mohammad.almamun@hsc.wvu.edu

## Abstract

### Background

Acute kidney injury (AKI) can lead to an approximate ninefold increased risk for developing chronic kidney disease (CKD). Despite this, many AKI survivors lack proper nephrology follow-up, highlighting the urgent need to identify patient profiles before onset CKD. Thus, we aimed to develop a patient profiling algorithm to identify clinical phenotypes from AKI to CKD progression.

### Methods

This retrospective study utilized electronic health records data from 2010 to 2022. We classified AKI into three groups: Hospital Acquired AKI (HA-AKI), Community Acquired AKI (CA-AKI), and No-AKI. We developed a custom patient profiling algorithm by combining network-based community and variable clustering methods to examine risk factors among three groups. The top three clusters were presented using comorbidities and medical procedures network graphs, and matched between two methods to find similarities and dissimilarities.

### Results

Among 58,876 CKD patients, 10.2% (5,981) and 11.5% (6,762) had HA-AKI and CA-AKI, respectively. The No-AKI group had a higher comorbidity burden compared to AKI groups, with average comorbidities of 2.84 vs. 2.04. Commonly risk factors observed in both AKI cohorts included long-term opiate analgesic use, atelectasis, history of ischemic heart disease, and lactic acidosis. The comorbidity network in

**Data availability statement:** The dataset could be requested to TriNetX upon an agreement with the requester. The data request link: https://trinetx.com/solutions/real-world-data-sets/#s_1 Contact email: Richard.Lilienthal@trinetx.com.

**Funding:** The author(s) received no specific funding for this work.

**Competing interests:** NO authors have competing interests.

HA-AKI patients was more complex compared to CA-AKI and No-AKI groups with higher number of diagnosis (64 vs 62 vs 55). The HA-AKI cohort had several conditions with higher degree (mean number of edges connected to each diagnosis) and betweenness centrality (bridges connecting different diagnosis clusters) including high cholesterol (34, 91.10), chronic pain (33, 103.38), tricuspid insufficiency (38, 113.37), osteoarthritis (34, 56.14), and removal of GI tract components (37, 68.66) compared to the CA-AKI cohort.

## Conclusion

Our proposed patient profiling algorithm successfully identifies AKI phenotypes toward CKD progression, offering a promising approach to identify early risk factors for CKD in improving targeted prevention strategies and reducing healthcare expenditures.

---

## 1. Introduction

It has been estimated that over 35.5 million U.S. adults have chronic kidney disease (CKD), yet 9 out of 10 are unaware of their disease [1,2]. Acute kidney injury (AKI) is independently associated with acute morbidity, mortality, and long-term kidney disease [3,4]. Hospital-acquired acute kidney injury (HA-AKI) is a heterogeneous syndrome and a common complication in acute care settings, especially among critically ill adults, up to 60% may experience it [5]. A recent study found that patients who underwent dialysis for HA-AKI should receive follow up care due to its association with reduced mortality and lower hospitalization rates [6]. However, less than 20% of HA-AKI survivors and only 21–50% of those who underwent acute dialysis within one year of hospitalization received nephrology follow-up [7,8]. In contrast, AKI acquired in the community, known as community-acquired AKI (CA-AKI) differs in risk factors, epidemiology, presentation, and impact compared to HA-AKI. A recent study of 734,340 hospital admissions reported that patients with HA-AKI had higher rates of in-hospital mortality (51.58% vs. 26.07%), longer average hospital stays (35.84±34.62 days vs. 21.25±22.35 days), and a greater need for dialysis during hospitalization (2.06% vs. 1.45%) compared to those with CA-AKI [9].

The association between the initial occurrence of AKI and long-term risk of CKD is multifactorial and complex [10]. Assessing the risk of AKI and its acceleration toward CKD or end-stage renal disease (ESRD) is a key area of research, with significant work focusing on its severity, recurrence, etiology, and clinical biomarkers [3,11]. Studies have shown that patients who experience AKI have an approximately nine-fold increased adjusted risk of developing CKD and a threefold increased adjusted risk of progressing to ESRD [12]. Additionally, a 2016 study estimated that the prevalence of recurrent AKI after a first episode could be as high as 25% [13]. AKI also contributes to a higher incidence of cardiovascular disease and progression to CKD [3,9,14–16]. However, identifying those at the highest risk, quantifying the extent of renal damage, and predicting the rate of disease progression remains challenging,

even with well-known associated comorbidities such as hypertension, diabetes mellitus, and pre-existing CKD. No study has fully explored the relationship between comorbidities, medical procedures following AKI, and long-term kidney injury. The transition from AKI to CKD carries a significant public health burden, increasing healthcare costs, hospitalizations, and mortality risk. Therefore, improving clinical phenotyping after an initial AKI event and understanding the patient's profile and trajectory toward CKD remains an area requiring further exploration and validation.

To better understand the multifactorial and complex progression, it is crucial to group clinical phenotypes and find the trajectory towards CKD which warrant a robust and comprehensive clustering methods. Traditional patient-centric clustering techniques, such as consensus clustering [17,18], K-means [19,20], latent class analysis [21], and hierarchical clustering [22], often overlook the underlying data structure and complex interactions among variables, as they primarily focus on grouping individuals based on observed similarities. To effectively identify different phenotypes, it is essential to group variables or clinical characteristics [23–25] based on inherent correlations and dependencies, which allows fora more granular understanding of data structure and the extraction of latent pattern. Furthermore, representing comorbidity relationships as a network can help characterize pathways of disease progression to CKD by quantifying and analyzing pairwise interactions among variables [26]. Current approaches are also limited in their ability to compare clinical phenotypes across multiple groups and are generally constrained by small, complex and multimodal datasets [23–27]. Moreover, recent machine learning prediction models did not also consider a patient's comprehensive phenotypic trajectory to predict AKI and its long-term outcomes [28–30].

To address these gaps, we propose a novel patient profiling framework that employs two complementary clustering methods: network-based and variable clustering. Identifying high-risk clinical phenotypes through a comprehensive patient profiling algorithm will enhance our understand of factors associated with the progression of AKI to CKD. Our study aims to develop and utilize two complementary clustering methods to understand combinations of risk factor driving AKI to CKD progression.

## 2. Methods

### 2.1 Study design and setting

This was a single center, STROBE-compliant retrospective observational study involving 90,602 patients with outpatient visits, covering inpatient, ambulatory, and emergency department encounters from February 2010 to June 2022. Given the retrospective nature of the data, informed consent was not required, as all patient information was de-identified prior to use. The study received an exemption from the Human Research Review Committee at the West Virginia University Institutional Review Board (IRB: # 2212689753).

### 2.2 Data sources

Data were obtained from the electronic health records (EHR) of a single Health Care Organization (HCO) – West Virginia using TriNetX. This HCO consisted of multi-hospital system within West Virginia. TriNetX is a global health research network that connects pharmaceutical companies, study sites, investigators and patients by sharing real-world data to facilitate clinical and observational research. The dataset typically includes information on diagnoses, procedures, encounters, medications, laboratory results, vital signs, genomic data, tumor properties, oncology treatments, tumor, chemotherapy lines, cohort details, and demographics.

### 2.3 Participants

Patients aged 18 years and older diagnosed with CKD using ICD-9-CM code 585 and ICD-10-CM code N18 were included in the study. These patients were followed retrospectively from the inception of the database until September 2022.

## 2.4 Definitions, inclusion, and exclusion criteria

Prior AKI events were identified using ICD-9-CM code 485 and ICD-10-CM code N17 within three years prior to their CKD diagnosis. Patients were followed after their first occurrence of AKI within those 3 years. Patients with any dialysis events before their CKD diagnosis prior to their first AKI diagnosis were excluded from the study. HA-AKI patients were defined as those who had AKI within 90-days of any inpatient hospitalization (identified by Current Procedural Terminology (CPT) codes: 99218–99239, 99251–99255, 99291, 99304–99307, 94002, G03378,1013699, and 1013659). CA-AKI patients were defined as those who had any AKI events, excluding HA-AKI, within three years prior their CKD diagnosis. After applying the inclusion and exclusion criteria, we created three cohorts: HA-AKI, CA-AKI, and No-AKI. A study design flow chart is shown in Fig 1.

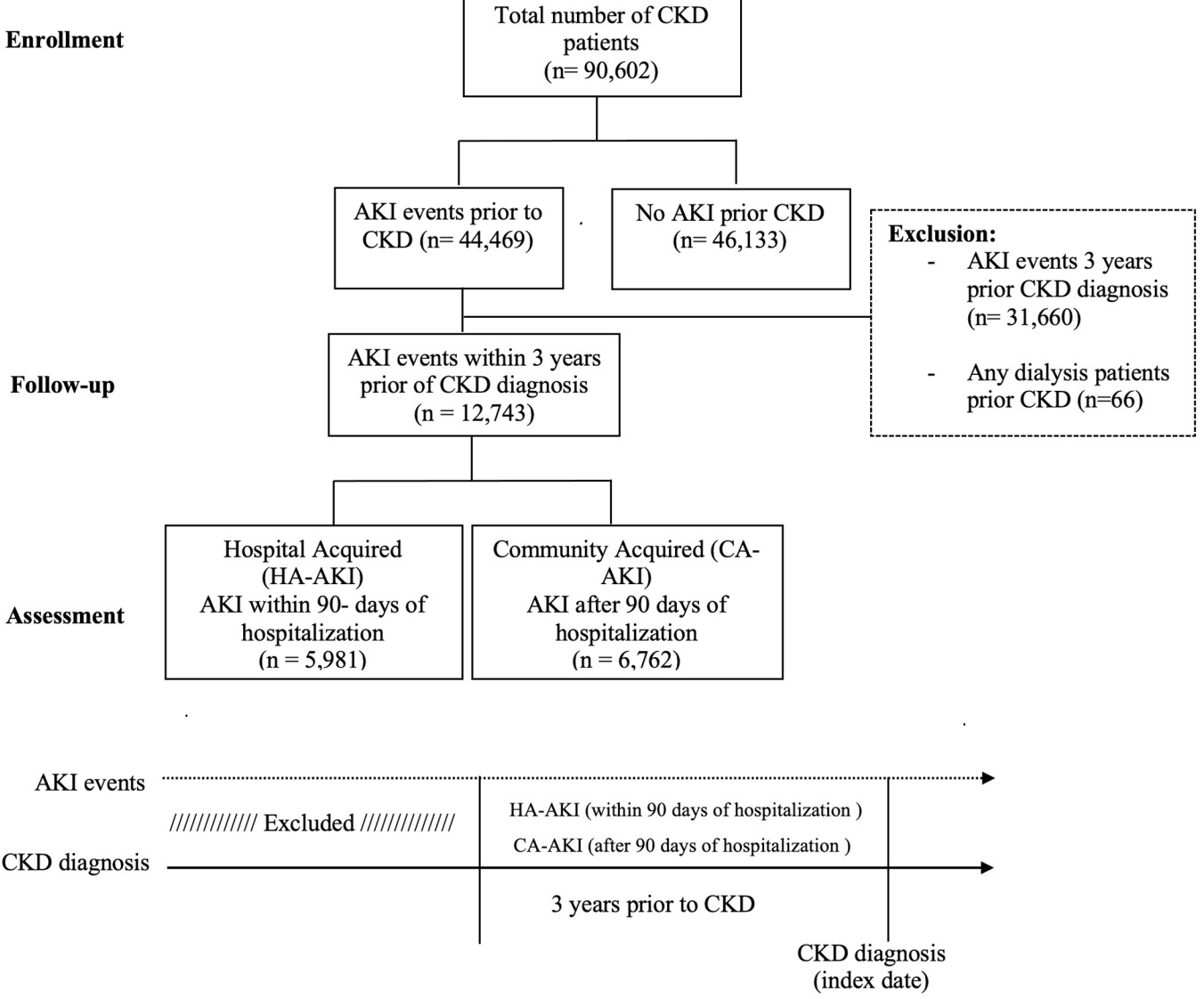

**Fig 1. Study design schema for the three study cohorts and their timelines.** Abbreviations: CKD-Chronic Kidney Disease, HA-AKI-Hospital acquired acute kidney injury, and CA-AKI-Community acquired acute kidney injury.

## 2.5  Identification of comorbidities and procedures

We examined and ranked all diagnoses and procedures codes for three cohorts. To obtain more reliable estimates, we excluded rarely diagnosed diseases and conducted procedures with a prevalence of less than 1% across all cohorts. Before ranking, we converted ICD-9-CM codes to ICD-10-CM codes to ensure consistency across cohorts. For the HA-AKI cohort, there were 64 diagnoses and 62 medical procedures codes. For the CA-AKI cohort, there were 62 diagnoses and 65 medical procedure codes. For No-AKI cohort, there were 55 diagnoses and 67 medical procedure codes.

## 2.6  Comorbidity network modeling

To understand the risk factor profiles associated with HA-AKI, CA-AKI, and non-AKI patients in relation to CKD, we constructed phenotypic disease and procedure network models. We binarized (i.e., 1 = presence, 0 = absence) all diagnoses and procedures for each cohort based on their presence or absence prior to CKD diagnosis. For procedures, we limited the analysis to those occurring within three years before CKD diagnosis. Phenotypic networks were then created for each cohort, with nodes representing disease diagnoses (comorbidities) and edges indicating co-occurrence relationships between pairs of comorbidities [26,31,32]. Network analysis offers a graphical representation of the complex patterns among risk factors. To quantify the strength of comorbidities between AKI to CKD and No-AKI to CKD groups, we introduced the observed-to-expected ratio (OER) [33]. The $OER_{ij}$, measures the strength of the comorbidity between disease pair $i$ and $j$, calculated as the ratio of the observed prevalence of the disease pair (O) to the expected prevalence (E), which is determined by the product of the prevalence of diseases $i$ and $j$:

$$OER_{ij} = \frac{C_{ij}N}{C_i C_j}$$

In this context, $C_{ij}$ represents the co-occurrence count of both diseases ($i$ and $j$), $N$ is the total number of patients in the population, and $C_i$ and $C_j$ are the prevalence of diseases ($i$ and $j$). The observed-to-expected ratio (OER) can be interpreted as a measure of relative risk. An OER greater than 1 indicates that the two diseases co-occur more frequently in the same patients than would be expected by chance, whereas an OER less than 1 suggests that the diseases are mutually exclusive. We included only those pairwise comorbidities in the network with a OERs greater than 1, as we aimed to identify risk factors specific to each cohort. To highlight prominent risk factors in the network graphs, we set the 90th percentile as the threshold for OER values in each cohort. For diagnoses, OER thresholds were 1.68, 1.84, and 2.25 for the HA-AKI, CA-AKI, and No-AKI cohorts, respectively. For procedures, the OER thresholds were 2.08, 2.14, 2.75 for the HA-AKI, CA-AKI, and No-AKI cohorts, respectively. To visualize the nodes clearly for each cohort, we have readjusted the edge weights and removed nodes with lower edge weights. The edge thresholding was done only for visualization purpose.

   To explore the complexity among the diagnoses and procedures, we incorporated five network metrics: diameter, degree centrality, betweenness centrality, average nearest neighbor path length, and closeness centrality [34]. Degree centrality represents the number of direct connections a node has with other diseases or procedures. An edge in the network signifies the interaction or relationship between nodes. The average degree is the mean number of edges connected to each node. The diameter of the network indicates the maximum number of edges that must be traversed to travel between the most distant nodes. Betweenness centrality measures the extent to which a node lies on the shortest paths between other pairs of nodes in the network. Closeness centrality is the reciprocal of the sum of the shortest path lengths between a node and all other nodes in a network. A higher closeness centrality value indicates that a disease is more likely to be diagnosed with other diseases in fewer steps. We used an unweighted edge to represent the presence or absence of relationships in a network, which means the network lacks temporal directionality for diseases and procedures. To identify cluster of closely related comorbidities, we employed the fast-unfolding cluster algorithm for community detection, which optimizes for the highest modularity score across different clustering layers [35,36]. This analysis allowed

us to identify the top three clusters of diagnoses and procedures for each cohort. Network analysis was performed using the *Gephi* network software package [37].

## 2.7 Hierarchical diagnosis and procedure clustering

To validate the community clustering of variables from the phenotypic network analysis, we employed a variable clustering method known as "*ClustOfVar*" [38]. This advanced algorithm is well-suited for datasets containing both quantitative and qualitative variables [38]. ClustOfVar provides a comprehensive understanding of data structure by grouping variables into clusters based on their similarities. It identifies groups of strongly related variables and constructs synthetic variables that summarize these clusters, thereby reducing data complexity while preserving key information. The primary metric used in *ClustOfVar* is the *homogeneity criterion*, which assesses how closely related the variables within a cluster are to a synthetic variable representing that cluster. The *ClustOfVar* algorithm offers a robust framework for variable clustering by maximizing homogeneity within clusters. It employs advanced statistical methods to handle mixed data types, ensuring a comprehensive understanding of the underlying data structures and facilitating the identification of meaningful patterns. Variable clustering uncovers correlated clinical and demographic variables, revealing underlying patterns that influence disease progression. In contrast, network-based clustering constructs networks of variables, offering both visual and analytical insights to identify and group critical comorbidities and procedures that drive the progression from AKI to CKD. These two approaches complement each other by cross-referencing identified clusters, ensuring robust and accurate phenotyping. The steps of the *ClustOfVar* algorithm are provided as a S1 File. We utilized ClustOfVar R package for running the cluster analysis from Chavent et al.[25].

## 2.8 Similarities and dissimilarities calculation

Each method identified top three clusters for diagnoses and procedures within each cohort. We then matched these clusters and calculated the similarity percentage among them. The final selection of diagnosis and procedure variables was based upon these matches. We anticipated that the hierarchical clustering algorithm would yield similarly complex networks of variables as those identified by the phenotypic disease network. Details of the algorithm steps used for comprehensive patient profiling and cluster matching between the two methods (see S1 File).

## 2.9 Statistical analysis

Descriptive statistics were employed to characterize the study population with continuous variables reported as means and standard deviations and categorical variables described using frequencies and proportions. Baseline characteristics of HA-AKI patient were compared with No-AKI patients, and CA-AKI were compared with No-AKI patients using Pearson chi-square tests for categorical variables and independent-samples t-tests for continuous variables. We performed regression analysis using cohort membership (i.e., HA-AKI, CA-AKI, or No-AKI) as the outcome and the selected comorbidities (binary) as predictors. The resulting odds ratios would demonstrate how specific components act as risk factors. All statistical analyses were performed using R (version 4.0.2, R Foundation for Statistical Computing, Vienna, Austria) [39].

## 3. Results

### 3.1 Demographic and clinical characteristics

A total of 58,876 patients were included in the analysis where 21.6% (12,743) had either HA- or CA- AKI and 78.4% (46,133) had No-AKI and (Fig 1). The sample predominantly comprised White race (84.02%), non-Hispanic or Latino (82.87%), females (52.03%) with a mean age of 61 years. Table 1 outlines the demographic characteristics of individuals with CKD who experienced AKI compared to those who did not. The AKI group had a higher proportion of White (88.7%) or non-Hispanic/Latino individuals (90.0%), a slightly higher percentage of females (51.4%), and a lower mean age 58.1

**Table 1. Baseline characteristics of the No Acute Kidney Injury (No-AKI) vs Hospital Acquired Acute Kidney Injury (HA-AKI) and No-AKI vs Community Acquired Acute Kidney Injury (CA-AKI) among the chronic kidney disease patients.**

| | AKI groups | | | | |
|---|---|---|---|---|---|
| | No-AKI | HA-AKI | p-value[1] | CA-AKI | p-value[2] |
| | N = 46133 | N = 5981 | | N = 6762 | |
| **Age** | | | <0.01 | | <0.01 |
| Mean Age (sd) | 68.6 (13.4) | 65.1 (13.6) | | 66.4 (13.2) | |
| **Age group (%)** | | | <0.01 | | <0.01 |
| <44 | 2429 (5.8) | 460 (8.2) | | 423 (6.7) | |
| 45-64 | 10707 (25.6) | 1994 (35.5) | | 2063 (32.6) | |
| >64 | 28662 (68.6) | 3161 (56.3) | | 3848 (60.8) | |
| **Sex (%)** | | | <0.01 | | <0.01 |
| Female | 24448 (53.0) | 2902 (48.5) | | 3284 (48.6) | |
| Male | 21671 (47.0) | 3077 (51.4) | | 3475 (51.4) | |
| Unknown | 14(0.03) | 2(0.03) | | 3(0.04) | |
| **Race (%)** | | | <0.01 | | <0.01 |
| White | 38172 (82.7) | 5322 (89.0) | | 5975 (88.4) | |
| Black | 1552 (3.4) | 206 (3.4) | | 287 (4.2) | |
| Other[3] | 106 (0.2) | 11 (0.2) | | 13 (0.2) | |
| Unknown | 6303 (13.7) | 442 (7.4) | | 487 (7.2) | |
| **Ethnicity (%)** | | | <0.01 | | <0.01 |
| Hispanic or Latino | 175 (0.38) | 17 (0.28) | | 23 (0.34) | |
| Not Hispanic or Latino | 37314 (80.9) | 5346 (89.4) | | 6128 (90.6) | |
| Unknown | 8644 (18.7) | 618 (10.3) | | 611 (9.04) | |
| **Comorbidities (%)** | | | | | |
| Anemia | 4648 (10.1) | 1651 (27.6) | <0.01 | 2181 (32.3) | <0.01 |
| Atrial Fibrillation | 3651 (7.9) | 187 (3.13) | <0.01 | 178 (2.6) | <0.01 |
| Anxiety | 8703 (18.9) | 981 (16.4) | <0.01 | 1244 (18.4) | <0.01 |
| Chronic obstructive pulmonary disease | 8203 (17.8) | 936 (15.6) | <0.01 | 1192 (17.6) | <0.01 |
| Heart Failure | 7425 (16.1) | 1192 (19.9) | <0.01 | 1500 (22.2) | <0.01 |
| Hyperlipidemia | 17424 (37.8) | 1182 (19.8) | <0.01 | 1567 (23.2) | <0.01 |
| Hypertension | 35184 (76.3) | 552 (9.2) | <0.01 | 972 (14.4) | <0.01 |
| Nicotine dependence | 15533 (33.7) | 865 (14.5) | <0.01 | 1262 (18.7) | <0.01 |
| Obesity | 9595 (20.8) | 899 (15.0) | <0.01 | 1308 (19.3) | <0.01 |
| Type 2 Diabetes Mellitus | 17919 (38.8) | 2625 (43.9) | <0.01 | 3146 (46.5) | <0.01 |

Abbreviations: HA-AKI: Hospital acquired acute kidney injury, CA-AKI: Community acquired acute kidney injury.

[1]pairwise statistical comparison for No-AKI vs HA-AKI;

[2]pairwise statistical comparison for No-AKI vs CA-AKI.

[3]Asian and Native Hawaiian or Other Pacific Islander are included as other race.

years) compared to the No-AKI group (61.6 years). The AKI group had a significantly lower mean age (HA-AKI: 65.8 ± 13.4 and CA-AKI: 66.4 (13.2) years) compared to the No-AKI group (68.6 ± 13.4 years). Additionally, the No-AKI group had a higher comorbidity burden, with a larger proportion having at least two comorbidities (mean number of comorbidities = 2.84 vs. 2.04; p < 0.05; 74.6% vs. 53.6%, respectively). Among the No-AKI group, 76.3% had hypertension, 33.7% had nicotine dependence, and 38.8% had Type 2 Diabetes Mellitus. In contrast, the CA-AKI group had a highest prevalence of heart failure (22.2%) and Type 2 Diabetes Mellitus (46.5%) and were significantly (p < 0.01) different than No-AKI group.

## 3.2. Network analytics of comorbidity and procedures

Fig 2 represents the visual network connections of study participants' diagnoses using 90th percentile OER values as the ranking factor to identify key nodes acting as critical 'bridges' between components within the network. Nodes with high 'betweenness' are strategic targets for each cohort. OER values greater than 1 were selected to highlight the strength of relationships between variables, with higher OERs indicating stronger associations. The edges were weighted, where a thicker edge indicates a stronger relationship between the two nodes. For visualization purpose, edge weight above 50th percentile was used to reduce the complexity of the network graph. The following values were taken as the 50th percentile: a) HA-AKI: 3.23 (0.13–6.32), b) CA-AKI: 4.31 (0.08–8.53), c) No-AKI:6.25 (0.07–12.43). Multiple nodes (ICD-9-CM/ ICD-10-CM codes) were identified in HA-AKI and CA-AKI cohorts, but not in no AKI cohort. Notably, the key nodes in HA-AKI and CA-AKI cohorts included: Z79.891 (long-term opiate analgesic use), J98.11 (atelectasis), Z82.49 (history of ischemic heart disease), E87.2 (lactic acidosis) and E11.65 (Type 2 diabetes mellitus with hyperglycemia). In contrast, the most significant nodes in the control group were I25.2, (previous myocardial infarction), Z90.49 (absence of other specified parts of digestive tract), and Z82.49 (family history of ischemic heart disease).

Edge weight > 50th percentile to reduce the complexity, HA-AKI: 3.23 (0.13–6.32), CA-AKI: 4.31 (0.08–8.53), No-AKI: 6.25 (0.07–12.43). Green, Purple, and Blue, representing different clusters of comorbidities observed among the three groups.

S1 Fig depicts the network of procedures encountered across the three cohorts (HA-AKI, CA-AKI, and No-AKI). Several nodes were common to both the HA-AKI and CA-AKI cohorts, including 84295 (sodium concentration measurement), 84540 (urea nitrogen measurement), 86923 (transfusion cross-matching donor blood compatibility), and 87070 (microbiology culture – non-blood specimen isolate). Unique to both the control and AKI patient networks were 82607 (vitamin B12measurement) and 83615 (lactate dehydrogenase measurement). Nodes specific to No-AKI cohort included 83605 (serum lactate measurement) and 85018 (hemoglobin measurement in whole blood samples).

Table 2 provides an overview of the network parameters for diagnoses and procedures across the three cohorts. While the complexity of the procedure networks was similar among the cohorts, the comorbidity network in HA-AKI patients was more complex compared to the No-AKI group, as evidenced by the higher number of nodes (64 vs. 55) and edges (645 vs. 520). The diagnosis network in the HA-AKI cohort was also more tightly clustered compared to the No-AKI cohort, as indicated by a significantly higher average degree (21.68 vs. 16.91, $p < 0.04$). For procedures, both AKI groups showed significantly higher measures for average degree, path length, betweenness and closeness compared to the No-AKI group.

## 3.3 Matched and similar diagnosis phenotypes

Our custom matching yielded the best alignment of clusters between both methods. S1 Table shows the percentage of clusters matched by both algorithms. For HA-AKI and CA-AKI, we achieved a 96% match among commodity phenotypes, while for the No-AKI cohort, the match was 100%. Table 3 presents the similar phenotypes found across the three cohorts. Among these cohorts, we identified 10 cardiovascular-related conditions, including hypercholesterolemia, atherosclerotic heart disease without angina pectoris, myocardial infarction, non-rheumatic mitral insufficiency, nonrheumatic tricuspid insufficiency, atrial fibrillation, heart failure, cardiomegaly, chest pain, and abnormal ECG/EKG findings. However, the *degree* and *betweenness* centrality differ across the cohorts.

Betweenness centrality measures the extent to which a node lies on the shortest paths between other nodes, indicating its influence within the network by controlling the flow of information. Diagnoses with high betweenness centrality act as "bridges" connecting different clusters or groups of diagnoses. These diagnoses may play a critical role in disease progression or serve as intermediaries in those pathways. In the HA-AKI cohort, conditions like chronic pain and osteoarthritis had higher degree and betweenness values (33, 103.38; 34, 56.14) compared to CA-AKI (25, 29.49; 26, 35.92) and no-AKI (27, 34.15; 27, 30.11) cohorts, respectively. When comparing HA-AKI to CA-AKI, several conditions exhibited higher betweenness centrality in the HA-AKI cohort, including high cholesterol (34, 91.10), chronic pain (33, 103.38),

A. HA-AKI Cohort

B. CA-AKI Cohort

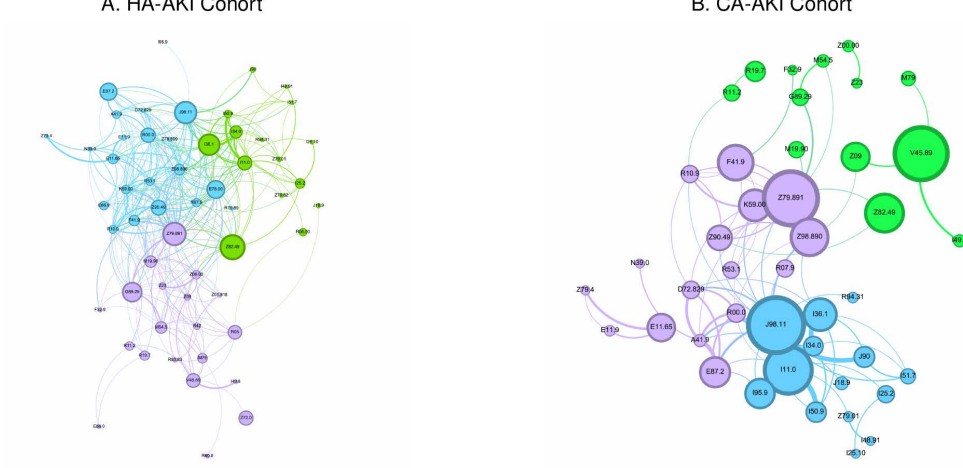

C. No-AKI Cohort

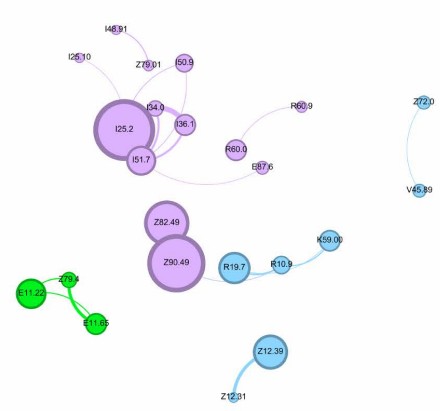

| Diagnosis | Description | Diagnosis | Description | Diagnosis | Description |
|---|---|---|---|---|---|
| A41.9 | Sepsis | I49.8 | Cardiac arrhythmias | R53.83 | Other fatigue |
| D72.829 | Elevated WBC count | I50.9 | Heart Failure | R60 | Edema |
| E11.22 | T2DM dibetic CKD | I51.7 | Cardiomegaly | R79.89 | Other abnormal blood chemistry |
| E11.65 | T2DM with hyperglycemia | I95.9 | Hypotension | R94.31 | Abnormal ECG EKG |
| E11.9 | T2DM without complications | J18.9 | Pneumonia organism | V45.89 | External hearingaid |
| E66.9 | Obesity | J90 | Pleural effusion | Z00.00 | Encounter general medical examination |
| E78.00 | Pure hypercholesterolemia | J98.11 | Atelectasis | Z01.818 | Encounter other preprocedural examination |
| E86.0 | Dehydration | K59.00 | Constipation | Z09 | Encounter other than malignant neoplasm |
| E87.2 | Acidosis | M19.90 | Osteoarthrosis | Z12.31 | Mammogram screening |
| E87.6 | Hypokalemia | M54.5 | Backache | Z12.39 | Breast cancer screening |
| F32.9 | Depressive disorder | M79 | Pain in limb | Z23 | Encounter for immunization |
| F41.9 | Anxiety disorder | N39.0 | Urinary tract infection | Z72.0 | Tobacco use disorder |
| G89.29 | Other chronic pain | R00.0 | Tachycardia | Z79.01 | Longterm anticoagulants use |
| I11.0 | Hypertensive heartdisease with HF | R06.00 | Dyspnea | Z79.4 | Longterm insulin use |
| I25.10 | Atherosclerotic heart disease | R07.9 | Chest pain | Z79.82 | Longterm aspirin use |
| I25.2 | Old myocardial infarction | R10.9 | Abdominal pain | Z79.891 | Longterm opiate analgesic use |
| I34.0 | Defect in mitral valve function | R11.2 | Nausea with vomiting | Z79.899 | Other long term (current) drug therapy |
| I36.1 | Nonrheumatic tricuspid insufficiency | R19.7 | Diarrhea | Z82.49 | Family history ischemic heart disease |
| I48.91 | Atrial Fibrilation | R42 | Dizziness giddiness | Z87.891 | Nicotine Dependence |
| I49.8 | Cardiac arrhythmias | R53.1 | Weakness | Z90.49 | Acquired absence other specified parts digestive tract |
| | | | | Z98.890 | Other specified post procedural states |

Edge weight > 50th percentile to reduce the complexity, HA-AKI: 3.23 (0.13-6.32), CA-AKI: 4.31 (0.08-8.53), No-AKI: 6.25 (0.07-12.43). Green, Purple, and Blue, representing different clusters of comorbidities observed among the three groups.

**Fig 2. The strongest associations of medical diagnosis across the three cohorts. a) HA-AKI, b) CA-AKI, and c) No-AKI.** The diagnosis clusters identified by the community detection algorithms and designated as three colors: green, purple, and blue. Each Node (e.g., Z82.49, Z79.891 etc.) represent diagnosis, node size indicates prevalence, while edge thickness (i.e., line between two nodes) represents the observed-to-expected ratio (OER) (> 90th percentile).

**Table 2. Network metrics in the comorbidity network of No Acute Kidney Injury (No-AKI) vs Hospital Acquired Acute Kidney Injury (HA-AKI) and No-AKI vs Community Acquired Acute Kidney Injury (CA-AKI) among the chronic kidney disease patients.**

| | Diagnosis | | | | | Procedure | | | | |
|---|---|---|---|---|---|---|---|---|---|---|
| | HA-AKI | | CA-AKI | | No-AKI | HA-AKI | | CA-AKI | | No-AKI |
| # patients | 5981 | | 6762 | | 46133 | 5981 | | 6762 | | 46133 |
| Nodes | 64 | | 62 | | 55 | 62 | | 65 | | 67 |
| Edges | 645 | | 651 | | 520 | 768 | | 814 | | 792 |
| Diameter[1] | 5 | | 5 | | 4 | 4 | | 5 | | |
| **Metrics** | | p-value | | p-value | | | p-value | | p-value | |
| Avg Degree[2] | 21.68 | 0.04 | 21.00 | 0.05 | 16.91 | 24.77 | 0.05 | 21.97 | <0.01 | 21.64 |
| Avg Path length[3] | 1.91 | N/A | 1.83 | N/A | 1.85 | 1.73 | N/A | 1.78 | N/A | 1.99 |
| Avg Betweenness[4] | 33.86 | 0.07 | 30.98 | 0.16 | 23.04 | 22.27 | <0.01 | 24.83 | <0.01 | 32.63 |
| Avg Closeness[5] | 0.53 | 0.17 | 0.55 | 0.22 | 0.56 | 0.67 | 0.12 | 0.58 | 0.26 | 0.53 |
| **Modularity** | | | | | | | | | | |
| Resolution | 1.00 | | 1.18 | | 1.34 | 1.21 | | 1.14 | | 1.11 |
| **Cluster distribution** | | | | | | | | | | |
| 1 | 41.18% | | 37.10% | | 49.09% | 48.39% | | 49.23% | | 56.72% |
| 2 | 32.35% | | 32.26% | | 40.00% | 38.71% | | 40.00% | | 22.39% |
| 3 | 26.47% | | 30.65% | | 10.91% | 12.90% | | 10.77% | | 20.90% |

p-values were calculated for pairwise comparison using t-test.

[1]Diameter: the maximum value of the weighted distance between any two nodes in the network;

[2]Avg. Degree: average number of links of all nodes in the network to other nodes;

[3]Avg path length: the average number of steps along the shortest paths for all possible pairs of network nodes;

[4]Avg Betweenness: how often a node is on the shortest path between other nodes in a network;

[5]Avg. Closeness: the average of the inverse of the shortest path lengths between the disease and all other diseases in the graph.

Abbreviations: AKI: Acute Kidney Injury, HA-AKI: Hospital acquired AKI, CA-AKI: Community acquired AKI. *P-value* represents comparison with the No-AKI group, N/A: The path length was very close to each other to determine the *p*-value

tricuspid insufficiency (38, 113.37), osteoarthritis (34, 56.14), general medical examination (32, 35.86), and removal of GI tract components (37, 68.66). Conversely, in the CA-AKI cohort, cardiomegaly (18, 14.83), abnormal ECG (20, 3.68), follow up exam (26, 47.11), and surgery (37, 74.25) showed higher betweenness values. When comparing either AKI cohort to the non-AKI group, the non-AKI cohort had higher betweenness centrality for conditions such as obesity (20, 52.71), myocardial infarction (31, 138.61), cardiomegaly (31, 53.45), abnormal chemistry findings (24, 6.30), follow up examinations (38, 96.47), and removal of components of the GI tract (37, 129.14).

### 3.4 Dissimilarities in diagnosis phenotypes

In comparing the dissimilarities among the three cohorts, we observed that sepsis (16, 26.31), tachycardia (29, 68.27), and dizziness/giddiness (23, 20.92) exhibited higher values in the HA-AKI cohort (Table 4). Conversely, in CA-AKI cohort, diagnoses such as type 2 diabetes with hyperglycemia (27, 172.36), hypoosmolality and hyponatremia (14, 22.00), hypertensive heart disease with heart failure (38, 99.68), COPD (unspecified) (11, 52.62), elevated WBC count (29, 22.26), and pleural effusion (24, 27.98) showed higher degree and betweenness compared to the HA-AKI cohort. Interestingly, anxiety disorder (A41.9) was the second-highest node in the HA-AKI cohort, with values of 36 and 53.95, respectively. It is noteworthy that pleural effusion (J90) had minimal impact on the phenotypic network of the CA-AKI cohort. Additionally, certain non-traditional factors, such as unspecified constipation, unspecified abdominal pain, and other external hearing aids, exhibited high degree and betweenness in both AKI cohorts but were not detected by the combined algorithms in the No-AKI cohort.

**Table 3. Comparison of comorbidities across the three cohorts. Similarities were determined by comparing the best-matched clusters identified by both algorithms. The table presents the degree and betweenness values for each comorbidity within the cohorts.**

| Similarity | HA-AKI (Degree[1], Betweenness[2]) | CA-AKI (Degree, Betweenness) | No-AKI (Degree, Betweenness) |
|---|---|---|---|
| **Endocrine, nutritional and metabolic diseases** | | | |
| T2DM without complications (E11.9) | (17,33.68) | (6, 3.56) | (4,0) |
| T2DM with hyperglycemia (E11.65) | (30,93.13) | (27, 172.36) | (15,31.93) |
| **Diseases of the circulatory system** | | | |
| Atherosclerotic heart disease without angina pectoris (I25.10) | (7,0.11) | (7, 0.00) | (6, 0.10) |
| Myocardial infarction (I25.2) | (18,36.84) | (18, 23.54) | (31, 138.61) |
| Nonrheumatic mitral insufficiency (I34.0) | (31,57.28) | (29, 27.86) | (24, 16.84) |
| Nonrheumatic tricuspid insufficiency (I36.1) | (38, 113.37) | (36, 61.50) | (28, 31.93) |
| Unspecified AFB (I48.91) | (5,0.00) | (8, 0.00) | (6, 0.00) |
| HF (I50.9) | (25,25.95) | (26, 29.91) | (21, 23.15) |
| **Medication usage** | | | |
| Long-term insulin use (Z79.4) | (3,0.00) | (3, 0.00) | (10,16.04) |
| Encounter for immunization (Z23) | (27,18.38) | (13, 1.96) | (7, 0.36) |
| Long term anticoagulants use (Z79.01) | (14,6.59) | (15, 3.27) | (15, 3.45) |
| Long term drug therapy (Z79.899) | (20,1.09) | (15, 0.10) | (2, 0.00) |
| **Medical Examinations and surgery** | | | |
| Encounter general medical examination without abnormal findings (Z00.00) | (32,35.86) | (13, 5.65) | (10, 1.71) |
| Encounter other preprocedural examination (Z01.818) | (14,3.83) | (12, 1.15) | (8, 2.06) |
| Follow-up examination for conditions other than malignant neoplasm (Z09) | (27,19.54) | (26, 47.11) | (38, 96.47) |
| Postprocedural removal of a components of the digestive tract (Z90.49) | (37,68.66) | (33, 41.00) | (37, 129.14) |
| Surgery (Z98.890) | (34,45.17) | (37, 74.25) | (27, 25.43) |
| **Others** | | | |
| Obesity (E66.9) | (22,29.88) | (19, 35.63) | (20, 52.71) |
| Other chronic pain (G89.29) | (33, 103.38) | (25, 29.49) | (27, 34.15) |
| Osteoarthritis unspecified site (M19.90) | (34,56.14) | (26, 35.92) | (27, 30.11) |

Abbreviations: No-AKI: No Acute Kidney Injury, HA-AKI: Hospital Acquired Acute Kidney Injury and CA-AKI: Community Acquired Acute Kidney Injury.

[1]Degree centrality represents the number of direct connections a node has with other diseases or procedures. The average degree is the mean number of edges connected to each node.

[2]Betweenness centrality measures the extent to which a node lies on the shortest paths between other nodes, indicating its influence within the network by controlling the flow of information.

S2 Table highlights the diagnoses which were not selected by either algorithm across the three cohorts, indicating their uniqueness to a specific AKI cohort. For example, in the HA-AKI cohort, distinct diagnoses included hypertension (I95.9*, 6,1), dyspnea unspecified (14,25.06), pneumonia (4,5.16), diarrhea (16,35.79), nausea with vomiting (14,31.74) and dehydration (2,0). In contrast, in the No-AKI cohort, unique findings were family history of IHD (38, 94.38), breast cancer screening (23, 67.08), mammogram screening (11, 0.85), screening malignant neoplasm colon (19, 10.43) which were not identified in the AKI cohorts.

### 3.5 Similarities in medical procedure phenotypes

S3 Table shows the similarities in procedures across the three cohorts. Basic metabolic and electrolyte panels, along with levels of medical care services, are common across all three cohorts with similar centrality and betweenness values. Liver function tests varied among the cohorts. Specifically, HA-AKI had higher degree and betweenness values

**Table 4. Dissimilarities of diagnosis among the cohorts.**

| Dissimilarity | HA-AKI (Degree, Betweenness) | CA-AKI (Degree, Betweenness) | No-AKI (Degree, Betweenness) |
|---|---|---|---|
| **Diseases of the circulatory system** | | | |
| Hypertensive heart disease with HF (I11.0) | (31,67.35) | (38, 99.68) | * |
| Cardiac arrhythmias (I49.8) | (7,1.84) | (9, 7.38) | * |
| Localized edema (R60.0) | (2,0.00) | (7, 2.84) | – |
| Tachycardia unspecified (R00.0) | (29,68.27) | (31, 24.67) | * |
| **Infection** | | | |
| Urinary tract infection, unspecified (N39.0) | (17,1.49) | (13, 0.00) | – |
| Sepsis (A41.9) | (16,26.31) | (23, 8.71) | * |
| **Electrolyte abnormalities** | | | |
| Hypoosmolality and hyponatremia (E87.1) | (2,0.00) | (14, 22.00) | * |
| Acidosis unspecified (E87.2) | (18,88.69) | (26, 53.61) | * |
| Hyperkalemia (E87.5) | (1,0.00) | (2, 0.00) | * |
| Hypokalemia (E87.6) | (1,0.00) | (13, 4.16) | – |
| **Diseases of the respiratory system** | | | |
| COPD unspecified (J44.9) | (6,17.27) | (11, 52.62) | – |
| Atelectasis (J98.11) | (32, 118.99) | (41, 119.66) | * |
| **Diseases of the musculoskeletal system** | | | |
| Lower back pain (M54.5) | (27,52.19) | (21, 16.11) | * |
| Soft tissue disorders (M79) | (18,39.02) | (17, 19.01) | – |
| Weakness (R53.1) | (31,25.42) | (26, 19.14) | – |
| Other fatigue (R53.83) | (16,4.67) | (8, 2.33) | – |
| **Mental and behavioral disorders** | | | |
| Major depressive episode first (F32.9) | (7,0.41) | (10, 1.40) | – |
| Dizziness and giddiness (R42) | (23,20.92) | (14, 9.72) | – |
| **Abnormal laboratory findings** | | | |
| Abnormal clinical and laboratory findings (R05) | (23,70.88) | (18, 31.90) | – |
| Elevated WBC count (D72.829) | (19,1.84) | (29, 22.26) | – |
| **Diseases of the gastrointestinal system** | | | |
| Constipation unspecified (K59.00) | (34,41.30) | (35, 59.39) | – |
| Unspecified abdominal pain (R10.9) | (31,34.48) | (27, 22.29) | – |
| **Other** | | | |
| External hearing aid (V45.89) | (24, 117.32) | (24, 128.72) | – |
| Long term aspirin use (Z79.82) | (10,1.39) | (5, 0.11) | – |

– Dissimilarities among AKI and No-AKI cohorts but were not identified in the community network clustering algorithm.

* Not available in the No-AKI dataset before clustering and network analysis.

for aspartate aminotransferase enzyme (33, 19.84) compared to the other cohorts. In the CA-AKI cohort, alanine aminotransferase and alkaline phosphatase had higher betweenness values (27, 86.72) and (31, 45.31), respectively. The coagulation factor prothrombin time (PT) exhibited higher degree and betweenness value (34, 64.01) in the non-AKI cohort compared to the other two cohorts. Among cardiovascular serum laboratory values, natriuretic peptide and quantitative troponin had higher degree and betweenness values (34, 41.77; 30, 10.32) in the HA-AKI cohort relative to the others. In the microbiology category, all components except blood aerobic with isolation showed higher degree and betweenness in the HA-AKI cohort. For radiology, all imaging studies demonstrated higher degree and betweenness in the HA-AKI cohort.

S4 Table showcases the likelihood of cohort membership for top 5 nodes for diagnoses from network analysis (i.e., the largest nodes). There were a few common diagnoses shared by cohorts. HA-AKI and CA-AKI had long-term use of an opiate analgesic (Z79.891) and Atelectasis (J98.11) in common, and HA-AKI and no AKI had family history of ischemic heart disease (Z82.49) in common. We found that J98.11 and nonrheumatic tricuspid valve insufficiency (I36.1) showed an increased risk of developing HA-AKI compared to those without these conditions (OR = 3.03 and 1.64, respectively, p < 0.001). hypertensive heart disease with heart failure (I11.0) and post femoral popliteal surgery (V45.89) had strong association to both HA-AKI (OR=3.15 and 1.20 respectively, p < 0.001) and CA-AKI (OR = 1.59 and 1.42 respectively, p < 0.001).

## 4. Discussion

### 4.1 Key findings

Our proposed comprehensive patient profiling algorithm identifies distinct clinical phenotypes in CKD patients with varying AKI histories. We observed that HA-AKI patients had a more complex comorbidity network than CA-AKI and No-AKI patients, with higher network centrality measures (e.g., degree and betweenness) for conditions such as high cholesterol, chronic pain, tricuspid insufficiency, and osteoarthritis. Diagnoses with high degree and between-ness indicates that they may play a pivotal role in linking otherwise unconnected diagnoses or cluster of diseases; i.e., they may be key intermediaries in the progression to CKD. This type of early recognition is crucial for estimating and predicting risk of CKD earlier than onset. Our network analysis corroborates known risk factors for AKI and CKD such as anemia, heart failure, and diabetes (Table 2) [40,41]. Our algorithm revealed an 80% similarity in clinical phenotypes between the AKI (HA-AKI and CA-AKI) and non-AKI cohorts. A significant finding of our study is the identification of non-traditional comorbidities and additional risk factors for the progression of AKI to CKD, such as mental and behavioral disorders (e.g., major depressive disorder) (Table 4). A 2021 population-matched cohort study of 30,998 patients with stress-related disorders (SRDs) found that patients with SRDs had an increased risk of AKI (Hazard Ratio: 1.22, 95% CI 1.04–1.42) and CKD progression (Hazard Ratio 1.23, 95% CI 1.10–1.37) [42]. Our study supports these findings by identifying less commonly recognized risk factors for AKI and CKD progression through a clustering model [42,43]. Although No-AKI group had a higher comorbidity burden, our network clustering analysis revealed greater complexity in the comorbidity networks of both AKI cohorts compared to No-AKI cohort. This suggests that clinicians should consider the combined burden of risk factors rather than focusing on individual factors when managing AKI patients.

Historically the risk of AKI and its progression to CKD has been associated with factors such as increasing age, Black race, male gender, and the presence of multiple comorbidities [44–46]. An observational study from 2021 suggests that the incidence of AKI increases after the age of 64 [47]. However, our findings indicate that the AKI cohorts had a lower mean age compared to the no-AKI cohort and were predominantly White and female (Table 1). This suggests that AKI is not restricted to the elderly or any specific race or gender. The literature supports that older patients are at higher risk for AKI due to age-related declines in renal function and a higher number of comorbidities [48,49]. Nonetheless, not all comorbidities contribute equally to the risk of AKI or CKD progression. Evaluating risk based solely on the number of comorbidities may not fully capture the associated risk. In our study, the non-AKI population exhibited a higher comorbidity burden compared to the AKI cohorts. One possible explanation is that patients with multiple chronic conditions may have been under closer clinical surveillance, leading to earlier CKD diagnosis and management, potentially reducing AKI risk. In contrast, AKI patients may have experienced more acute insults (e.g., hospitalization, sepsis, nephrotoxic exposure) rather than a higher baseline burden of chronic diseases. These findings suggest that assessing the risk of AKI and its progression to CKD requires a comprehensive examination of various demographic and clinical factors to accurately identify specific risk factors and clinical phenotype hazards.

## 4.2 Network analysis comparison across three cohorts

In our network analysis across the three cohorts, the AKI cohort exhibited higher quantitative values for edges and nodes compared to the non-AKI cohort, indicating a greater number of connections between nodes (Table 3). This analysis also revealed higher averages for degree and betweenness within the AKI cohort, particularly highlighting the overlap of comorbidities and demonstrating more closely clustered variables compared to the non-AKI cohort. Clinically, this suggests that the AKI cohort presents greater complexity, necessitating a more comprehensive examination of factors when assessing individuals at risk for CKD. Comorbidities shared across all cohorts were predominantly cardiac in nature, such as atrial fibrillation, heart failure, and atherosclerotic heart disease (Table 2). Conversely, sepsis, a well-known cause of AKI in critically ill patients and a major contributor to ICU morbidity and mortality, was highlighted. A 2011 study found that 40% of critically ill patients develop sepsis following AKI, indicating that AKI may increase the risk of sepsis [50]. The challenge of determining which syndrome occurs first is notable, as both conditions are interrelated. Our findings align with previous research, particularly in comparing HA-AKI to CA-AKI cohorts. We observed that sepsis had higher betweenness in the HA-AKI cohort (26.31 vs. 8.71), underscoring its significant association with AKI risk (Table 4). Additionally, diagnoses leading to intravascular volume depletion, such as diarrhea, nausea with vomiting, and dehydration, were uniquely associated with the HA-AKI cohort. The association between HA-AKI and volume depletion diagnoses (e.g., diarrhea, nausea with vomiting, dehydration) may be partially influenced by differences in data availability. These conditions are more easily captured in hospitalized patients, where fluid status is closely monitored, whereas outpatient volume depletion is likely underreported or inconsistently documented in EHR data. Future research with comprehensive outpatient laboratory and clinical data is needed to confirm these findings.

## 4.3 Similarities and dissimilarities of diagnoses and procedures in three cohorts

The cluster matching and similarity analysis confirms the most prominent risk factors for AKI such as T2DM (E11.9 and E11.65), Myocardial infarction (I25.2), and heart failure (I50.9) [51–53]. Moreover, the regression analysis after filtering top 5 comorbidities corroborates with the current findings such as association between AKI and long-term pain medications, Atelectasis, and nonrheumatic tricuspid valve insufficiency [54,55]. In addition, AKI cohorts had higher degrees and betweenness for procedures related to basic metabolic and electrolyte panels. This is consistent with expectations, as patients with kidney disease typically require closer monitoring of serum electrolyte levels [49]. In the HA-AKI cohort, aspartate aminotransferase enzyme measurements showed higher degrees and betweenness compared to the other cohorts. While the development of CKD post-AKI and its associated risk factors are well-documented in patients without cirrhosis [56], our study did not include patients with cirrhosis. The increased degree and betweenness of aspartate aminotransferase in the HA-AKI cohort may suggest potential hepatic involvement, despite the absence of cirrhosis. Conversely, in the non-AKI cohort, cardiovascular serum laboratory values such as natriuretic peptide and quantitative troponin levels exhibited higher degrees and betweenness (34, 41.77; 30, 10.32) compared to the AKI cohorts (13, 4.71; 11, 0.91 and 13, 3.42; 6, 0.37, respectively). This finding is contrary to expectations, given the established link between AKI and cardiovascular disease [57]. This discrepancy highlights an area that warrants further investigation to reconcile these unexpected results.

## 4.4 Combining clustering and network-based methods

Unlike methods that focus solely on grouping individuals, ClustOfVar analyzes the data structure by clustering variables based on their intrinsic relationships. It accommodates both quantitative and qualitative variables, creating synthetic variables to reduce complexity while preserving essential information. ClustOfVar enhances intra-cluster cohesion through a homogeneity criterion, a feature not commonly emphasized in other methods. This approach provides a more nuanced understanding of data complexity, uncovering hidden patterns and dependencies that might be missed in patient-level

analyses. The hierarchical clustering routine and its dendrogram facilitate visual interpretation and the determination of the optimal number of clusters, making it a robust tool for analyzing diverse and complex datasets.

### 4.5 Strengths and implications of the proposed method

Our innovative comprehensive patient profiling tool accurately projects a patient's progression from AKI to CKD using diagnosis and procedures data. This custom profiling algorithm integrates two clustering methods (ClustOfVar and network-based community clustering), thereby enhancing the validity of traditional clustering approaches. Our network analysis not only quantifies the number of comorbidities and procedures associated with AKI to CKD progression but also identifies specific types of comorbidities and procedures relevant to the patient's journey. This tool confirms established risk factors for AKI to CKD progression, such as hypertension and diabetes, and further highlights non-traditional risk factors, including alcohol use disorder and characteristics specific to the non-AKI group. Cautions should be taken to generalize and apply these findings directly towards predicting AKI, but our custom algorithm could be used as an "apriori" to filter potential risk factors (e.g., traditional and non-traditional) for understanding the trajectory of a patient health. It also addresses the knowledge gap of cluster analysis towards multimodal clinical data for understanding the heterogeneity in disease populations [27,58].This tool can be applied to understand the progression of other chronic diseases (e.g., heart failure, COPD, diabetes). It is capable to identify both similar and distinct risk factor clusters among patient groups. Unlike previous studies that have focused on pre-selected risk factors (e.g., diabetes and hypertension), our tool provides a comprehensive list of otherwise unidentified risk factors affecting AKI to CKD progression. Additionally, it can be used to assess the long-term impact of chronic diseases, particularly by linking the clusters revealed in both clustering and network analyses to demographic factors such as age, gender, race, and ethnicity.

### 4.6. Limitations

Our study has several limitations that should be considered. First, the retrospective design introduces the risk of missing data, which could lead to confounding bias. For instance, AKI and CKD patients were identified using ICD-9-CM and ICD-10-CM codes due to the absence of serum creatinine values. Second, our definition of CA-AKI was based on inclusion and exclusion criteria, as we could not measure prior exposure to CA-AKI. Third, the results may be affected by residual biases and unmeasured confounders, as some commonly associated conditions (e.g., diseases or procedures with <5% prevalence) were excluded. Fourth, our algorithm does not capture temporal relationships between comorbidities and procedures related to AKI progression to CKD, largely due to gaps in continuous healthcare insurance coverage. Fifth, the caution should be taken to interpret the baseline characteristics for other races such as Black, Asian, as our dataset consisted has higher percentage of population with White race.

## 5. Conclusion

Our custom comprehensive patient profiling algorithm offers a novel method for grouping and identifying clinical phenotypes from AKI to CKD progression to successfully identify the traditional and non-traditional risk factors. This method is scalable and adaptable to similar chronic diseases to improve risk characterization form multimodal and heterogenous clinical data, which may enhance clinical decision-making. This approach holds promise for advancing a deeper understanding of various AKI phenotypes and addressing the clinical gap leading to CKD. Future research should include prospective, multi-center studies to evaluate the impact of different stages and durations of AKI on the risk of long-term CKD complications. Additionally, exploring temporal patterns of comorbidities and procedures in relation to other chronic diseases using Medicaid and private healthcare claims data could be beneficial. This tool also has the potential to uncover non-traditional risk factors for chronic diseases by identifying less prevalent or previously unknown comorbidities and procedures.

## Supporting information

**S1 Fig. The strongest associations of medical procedure for HA-AKI, CA-AKI, and No-AKI cohort.** The procedure clusters identified by the community detection algorithms and designated as three colors: green, purple, and blue. Each Node (e.g., 84295, 87070, 84540 etc.) represent procedures, node size indicates prevalence, while edge thickness (i.e., line between two nodes) represents the observed-to-expected ratio (OER) (> 90th percentile).
(TIFF)

**S1 Table. Unique phenotypes which were not matched by either algorithm.**
(XLSX)

**S2 Table. Diagnoses which were not selected by either algorithm across the three cohorts.**
(XLSX)

**S3 Table. Similarities of the procedures among the cohorts.**
(XLSX)

**S4 Table. Regression analysis of the top nodes identified by both algorithms.**
(XLSX)

**S1 File. Supplementary text.**
(DOCX)

## Author contributions

**Conceptualization:** Mohammad A. Al-Mamun.

**Data curation:** Mohammad A. Al-Mamun, Ki Jin Jeun, Imtiaz Ahmed.

**Formal analysis:** Mohammad A. Al-Mamun, Imtiaz Ahmed.

**Investigation:** Todd Brothers, Ernest O. Asare.

**Methodology:** Mohammad A. Al-Mamun, Imtiaz Ahmed.

**Project administration:** Mohammad A. Al-Mamun.

**Software:** Ki Jin Jeun.

**Supervision:** Mohammad A. Al-Mamun.

**Validation:** Mohammad A. Al-Mamun, Todd Brothers, Khaled Shawwa.

**Visualization:** Ki Jin Jeun, Imtiaz Ahmed.

**Writing – original draft:** Mohammad A. Al-Mamun.

**Writing – review & editing:** Mohammad A. Al-Mamun, Todd Brothers, Ernest O. Asare, Khaled Shawwa, Imtiaz Ahmed.

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
