## [Decision Letter · Decision Letter 0]

PONE-D-24-34699Evaluating the kidney disease progression using a comprehensive patient profiling algorithm: A hybrid clustering approachPLOS ONE

Dear Dr. Al-Mamun,

Thank you for submitting your manuscript to PLOS ONE. After careful consideration, we feel that it has merit but does not fully meet PLOS ONE’s publication criteria as it currently stands. Therefore, we invite you to submit a revised version of the manuscript that addresses the points raised during the review process.

**ACADEMIC EDITOR: **1. It's really complicated how you describe creating a unique phenotypic disease and procedure network. Provide readers with a more thorough workflow diagram or pseudocode to assist them comprehend the stages involved in the development of this method.2. You made reference to utilizing information from electronic health records (EHRs), but you did not elaborate on how you dealt with outliers or missing data. Giving specifics on these elements would increase the validity of your research findings.3. Although you described the classifications used for AKI categories, it would be beneficial to have additional information on the standards used. For example, describe the criteria used to define "Community Acquired AKI" versus "Hospital Acquired AKI."4. Although the IRB approval, of the study is commendable, you should talk about any ethical issues with using patient data, even if it was de-identified. This would increase your study's credibility.5. You made many references to supplemental tables and figures. To ensure narrative coherence, make sure these are easily readable and labeled, and that crucial information is condensed into the main body of the text.. 

We look forward to receiving your revised manuscript.

Kind regards,

Wisit Kaewput, MD

Academic Editor

PLOS ONE

Journal requirements:    When submitting your revision, we need you to address these additional requirements. 1. Please ensure that your manuscript meets PLOS ONE's style requirements, including those for file naming. The PLOS ONE style templates can be found at https://journals.plos.org/plosone/s/file?id=wjVg/PLOSOne_formatting_sample_main_body.pdf and https://journals.plos.org/plosone/s/file?id=ba62/PLOSOne_formatting_sample_title_authors_affiliations.pdf 2. Please note that PLOS ONE has specific guidelines on code sharing for submissions in which author-generated code underpins the findings in the manuscript. In these cases, we expect all author-generated code to be made available without restrictions upon publication of the work. Please review our guidelines at https://journals.plos.org/plosone/s/materials-and-software-sharing#loc-sharing-code and ensure that your code is shared in a way that follows best practice and facilitates reproducibility and reuse. 3. We note that you have indicated that there are restrictions to data sharing for this study. For studies involving human research participant data or other sensitive data, we encourage authors to share de-identified or anonymized data. However, when data cannot be publicly shared for ethical reasons, we allow authors to make their data sets available upon request. For information on unacceptable data access restrictions, please see http://journals.plos.org/plosone/s/data-availability#loc-unacceptable-data-access-restrictions.  Before we proceed with your manuscript, please address the following prompts: a) If there are ethical or legal restrictions on sharing a de-identified data set, please explain them in detail (e.g., data contain potentially identifying or sensitive patient information, data are owned by a third-party organization, etc.) and who has imposed them (e.g., a Research Ethics Committee or Institutional Review Board, etc.). Please also provide contact information for a data access committee, ethics committee, or other institutional body to which data requests may be sent. b) If there are no restrictions, please upload the minimal anonymized data set necessary to replicate your study findings to a stable, public repository and provide us with the relevant URLs, DOIs, or accession numbers. Please see http://www.bmj.com/content/340/bmj.c181.long for guidelines on how to de-identify and prepare clinical data for publication. For a list of recommended repositories, please see https://journals.plos.org/plosone/s/recommended-repositories. You also have the option of uploading the data as Supporting Information files, but we would recommend depositing data directly to a data repository if possible. Please update your Data Availability statement in the submission form accordingly. 4. Please include captions for your Supporting Information files at the end of your manuscript, and update any in-text citations to match accordingly. Please see our Supporting Information guidelines for more information: http://journals.plos.org/plosone/s/supporting-information.  5. We notice that your supplementary figures are uploaded with the file type 'Figure'. Please amend the file type to 'Supporting Information'. Please ensure that each Supporting Information file has a legend listed in the manuscript after the references list.

Additional Editor Comments:

1. It's really complicated how you describe creating a unique phenotypic disease and procedure network. Provide readers with a more thorough workflow diagram or pseudocode to assist them comprehend the stages involved in the development of this method.

2. You made reference to utilizing information from electronic health records (EHRs), but you did not elaborate on how you dealt with outliers or missing data. Giving specifics on these elements would increase the validity of your research findings.

3. Although you described the classifications used for AKI categories, it would be beneficial to have additional information on the standards used. For example, describe the criteria used to define "Community Acquired AKI" versus "Hospital Acquired AKI."

4. Although the IRB approval, of the study is commendable, you should talk about any ethical issues with using patient data, even if it was de-identified. This would increase your study's credibility.

5. You made many references to supplemental tables and figures. To ensure narrative coherence, make sure these are easily readable and labeled, and that crucial information is condensed into the main body of the text.

Reviewers' comments:

Reviewer's Responses to Questions

**Comments to the Author**

1. Is the manuscript technically sound, and do the data support the conclusions?

Reviewer #1: Partly

Reviewer #2: Yes

Reviewer #3: No

Reviewer #4: Yes

Reviewer #5: Partly

Reviewer #6: Yes

2. Has the statistical analysis been performed appropriately and rigorously? 

Reviewer #1: Yes

Reviewer #2: Yes

Reviewer #3: Yes

Reviewer #4: Yes

Reviewer #5: Yes

Reviewer #6: Yes

3. Have the authors made all data underlying the findings in their manuscript fully available?

Reviewer #1: Yes

Reviewer #2: Yes

Reviewer #3: No

Reviewer #4: Yes

Reviewer #5: Yes

Reviewer #6: Yes

4. Is the manuscript presented in an intelligible fashion and written in standard English?

Reviewer #1: No

Reviewer #2: Yes

Reviewer #3: Yes

Reviewer #4: Yes

Reviewer #5: Yes

Reviewer #6: Yes

5. Review Comments to the Author

Reviewer #1: The manuscript from Al-Mamun et al aims to characterise the clinical phenotypes of AKI and/or CKD patients, using different clustering and network algorithm on a large electronic health records dataset. Patients with a first diagnosis of CKD were selected and the presence or absence of AKI in the previous three years was characterised. Then, network and clustering of medical records were performed in “hospital acquired AKI”, “community acquired AKI” or “no previous AKI” patients with CKD. The main result is unclear. Despite appropriate method description, it is not clear how to interpret data. The multiplicity of tables, figures, analyses, supplementary files, specific/unusual metrics (with limited commentary) and the overall mixing in figure numbering, missing figure labels and cryptic coded labelling further complicates understanding.

Regarding the study design, it is not very clear what healthcare is provided by WVU, which is highly relevant to understand the selection of patients. Are patients hospitalised or having medical visits? It would also be very interesting to characterise the patient trajectories as in the number of hospital/medical visits during the 3 years prior CKD, and at least eGFR and albuminuria levels observed at CKD diagnosis if available.

The paper could be improved by reducing the number of tables/figure, while improving their quality/readability. Many diagnoses/procedures remain coded by ICD or CPT codes, which limits interpretations. A schematic of representation of the network terms/items and metrics and their interpretation could enhance understanding.

- Table 1 is confusing as it is not stated that these are actually CKD patients, with a history of AKI or not. Also, the table data is somewhat redundant (NO AKI vs AKI results and p-values could be removed).

- Figure 2a and 2b would be greatly improved by ordering items by the most relevant metrics, such as centrality in No-AKI and labelling the different diagnosis. Cohorts names in the text should be corrected.

- Table 3 provides similarity values, but it is unclear how to interpret the results, and the associated paragraph mainly repeats observation rather than

- What softwares were used? Was it R? any other?

- Where are figure legends?

- There are MANY issues in figure numbering

Reviewer #2: This study uses an innovative comprehensive patient profiling tool accurately projects a patient’s progression from AKI to CKD using diagnosis and procedures data. It demonstrates how different comorbidities and procedures can be analyzed using a profiling method to elucidate factors influencing the transition from AKI to CKD. It identifies both similar and distinct risk factor clusters among patient groups and has the potential to uncover non-traditional risk factors for chronic diseases by identifying less prevalent or previously unknown comorbidities and procedures.

However, as described in the Limitations of the study, the retrospective design introduces the risk of missing data, which could lead to confounding bias, and should be better discussed in the manuscript as described below:

1) In Table one, “race and “ethnicity” have a high percentage of “unknown” data. As in CKD we know the importance of this , it should better explored and discussed (why is only “white” and “non white” ?, for example). In discussion is said that “the risk of AKI and its progression to CKD has been associated with factors such as increasing age, Black race, male gender, and the presence of multiple comorbidities.” Why there is no information about black race in this study, only “non -white” ?

2) In results , it is said that “The No-AKI group had a higher comorbidity burden compared to AKI group”. What is the reason for that ? Is there any hypothesis?

3) Finally, in discussion, intem 4.2, 4.2: “Additionally, diagnoses leading to intravascular volume depletion, such as diarrhea, nausea with vomiting, and dehydration, were uniquely associated with the HA-AKI cohort.” This fact couldn´t be only due the lack of information, as it is easier to have this data when the patient is in hospital ?

Reviewer #3: This manuscript is interesting; however, several edits are necessary to improve the research article:

1. The background section needs better organization, particularly in discussing AKI versus HA-AKI.

2. I recommend relocating the following passage to the methods section: “Variable clustering uncovers correlated clinical and demographic variables, revealing underlying patterns that influence disease progression. In contrast, network-based clustering constructs networks of variables, offering both visual and analytical insights to identify and group critical comorbidities and procedures that drive the progression from AKI to CKD. These two approaches complement each other by cross-referencing identified clusters, ensuring robust and accurate phenotyping.”

3. While the study asserts as STROBE-compliant, I suggest providing supporting evidence for this.

4. I find a mismatch between the study's stated aim and the actual work conducted. The authors assert that the aim is to understand the progression from AKI to CKD, yet the findings seem to focus on identifying risk profiles associated with CKD patients with and without AKI. This is further emphasized in the discussion section. For clarity and consistency, I recommend aligning the objective with what was accomplished in the study.

Reviewer #4: This research article is a commendable piece of scholarly work that contributes significantly to its field. The methodology adopted is robust, ensuring the reliability and validity of the findings. The literature review is comprehensive and provides a solid foundation for the research. The analysis is insightful, revealing new aspects that were previously unexplored. The conclusions drawn are well-supported by the data, and the implications for future research are thought-provoking. The authors have done an excellent job of communicating complex ideas in a clear and accessible manner, making the article a valuable resource for both experts and novices. Overall, this article is a fine example of academic rigor and intellectual acumen, and it is sure to inspire further inquiry and discussion. Kudos to the authors for their hard work and dedication to advancing knowledge in their field.

Reviewer #5: In this paper, Mohammad A. AI-Mamun and colleagues developed a new phenotyping framework for hybrid clustering (combine variable clustering and network-based clustering) to investigate the main risk factors that lead to chronic kidney disease (CKD). Specifically, the authors explored the risk of CKD after an acute-kidney injury (AKI) addressing multifactorial and complex progression from AKI to CDK, whith the goal of improving clinical decision and patient care.

I believe the core idea of this study is innovative. However, several points are unclear, making it difficult to follow the authors’ rationale and the results they presented. Consequently, the manuscript needs a deep revision. Below are my detailed comments:

ABSTRACT:

o The authors could add a bit more context to explain why understanding the AKI-to-CDK transition is so crucial, particularly in terms of public health impact.

o I suggest to report the number of patients used for the analysis after the filtering (58,606 patients) instead of the total dataset (90,602).

INTRODUCTION:

o I strongly recommend discussing in more depth the limitations of existing clustering methods, while highlighting the advantanged of the authors’ proposed approach.In particular, the authors could explain why traditional patient-centric clustering techniques are inadequate for understanding this problem (highlighting their inability to capture temporal or interaction complexities).

o It might also help to mention upfront how your two complementary methods (variable clustering and network-based clustering) were combined.

MATERIAL AND METHOD:

o Please, define “ICD-9-CM” and “ICD-10-CM” (perhaps in “Definitions, Inclusion, and Exclusion criteria” section) for clarity. In addition, ensure consistency in how these are referenced throughout the text (for example “ICD-9-CM” or “ICD-CM 9”).

o Since the authors refer to the No-AKI cohort both as “No-AKI cohort” and “cohort 3”, please clearly define the cohorts when introducing them. For example: “After applying the inclusion and exclusion criteria, we created three cohorts: HA-AKI (cohort 1), CA-AKI (cohort 2), and No-AKI (cohort 3).”

o The language used in the “Statistical Analysis” section could be revised for improve readability.

o Specify the statistical method used to compare network measures between AKI groups and No-AKI reference group.

o In the sentence, “We binarized all diagnoses and procedures for each cohort based on their presence or absence prior to CKD diagnosis.”, it would be useful to clarify if 1 was used for presence and 0 for absence, or viceversa.

o The process of developing the phenotipic network need clarification: Do the edges link diagnoses of disease recorded only whithin the same admission? From the cited literature, it seems edges link subsequent diagnoses of disease. Please explain this aspect in more detail. I highly recommend adding a Supplementary Figure illustrating the main components of the network.

o When discussing the observed-to-expected ratio (OER), consider explaining in simple terms what it tells us—perhaps with a short example illustrating how co-occurrence affects the network.

o The authors generated networks with undirected edges. Since the dataset spans multiple years (2010-2016), it is likely that certain diseases and/or procedures are consequences of others. Consider introducing temporal analysis to study how comorbidities and procedures evolve over time in AKI to CKD progression. The authors could model the sequence of disease occurrence and procedure over time, rather than treating them as static variables.

o A survival analysis (e.g., Kaplan-Maier curves) could be used to estitate the time from AKI to CDK and to identify time-dependent risk factors.

RESULTS:

o In the final part of the introduction, the authors mention that one of the aim of this study is to identify disease-specific trajectories comparing HA-AKI and CA-AKI subpopulations. I recommend adding two additional columns to Table 1 to show the p-values for comparisons between HA-AKI and No-AKI, and between CA-AKI and No-AKI. Consequently, the main text should discuss the key differences between HA-AKI and CA-AKI.

o Please check whether the figures are cited correctly in the text. For instance, Figure 2a seems to show degree measures, while Figure 2b shows betweenness centrality. However, the text refers only to Figure 2a, describing it as displaying betweenness centrality. Additionally, the caption for Figure 2 in the main text might be incorrect.

o In supplementary figure 1, what do the colorsrepresent? Please add a legend or include this information in the text. In addition, insert that “Supplementary Figure 1A” refers to HA-AKI and so on.

o In Supplementary Figure 1a, some edges appear thicker (for example, the link between nodes 82728 and 83540 in green). Are the edges weighted?

o Since the authors commented in the main text the common nodes between the two AKI subtypes, I strongly suggest conducting a graph subtraction analysis. The authors could remove edges common to both networks, retaining only disease-specific paths.

o What is the Supplementary Figure 2a? Please, clarify.

o Ensure that Figure 3 is cited correctly in the main text.

o The figures displaying the networks are unclear. I recommend either finding an alternative way to present the data or reducing the number of links to improve clarity.

o After using the network to filter comorbidities and procedures, the authors could perform regression analysis using cohort membership (i.e., HA-AKI, CA-AKI, or No-AKI) as the outcome and the selected comorbidities or procedures (binary) as predictors. The resulting odds ratios would demonstrate how specific components act as risk factors.

DISCUSSION: I suggest including references to studies that have used similar methodologies and explaining the advantages of the approach used here.

Reviewer #6: This is a novel study with relevant findings that in some way question the existing assumptions regarding comorbidities that predispose to chronic kidney disease after acute kidney injury. The limitations of the study are well stated and the designs proposed to delve deeper into the subject are interesting.

6. PLOS authors have the option to publish the peer review history of their article (what does this mean? ). If published, this will include your full peer review and any attached files.

**Do you want your identity to be public for this peer review?** For information about this choice, including consent withdrawal, please see our Privacy Policy .

Reviewer #1: No

Reviewer #2: **Yes: ** Patricia Maluf Cury

Reviewer #3: No

Reviewer #4: **Yes: ** Tabeer Tanwir Awan

Reviewer #5: No

Reviewer #6: No

---

## [Author Response · Author response to Decision Letter 1]

2 Apr 2025

We have attached the response to the reviewer as a seprate file. Also, we have a cover letter to the Editor.

---

## [Decision Letter · Decision Letter 1]

Evaluating the kidney disease progression using a comprehensive patient profiling algorithm: A hybrid clustering approach

PONE-D-24-34699R1

Dear Dr. Al-Mamun,

We’re pleased to inform you that your manuscript has been judged scientifically suitable for publication and will be formally accepted for publication once it meets all outstanding technical requirements.

Kind regards,

Wisit Kaewput, MD

Academic Editor

PLOS ONE

Additional Editor Comments (optional):

Accept as is.

Reviewers' comments:

Reviewer's Responses to Questions

**Comments to the Author**

1. If the authors have adequately addressed your comments raised in a previous round of review and you feel that this manuscript is now acceptable for publication, you may indicate that here to bypass the “Comments to the Author” section, enter your conflict of interest statement in the “Confidential to Editor” section, and submit your "Accept" recommendation.

Reviewer #5: All comments have been addressed

Reviewer #6: All comments have been addressed

2. Is the manuscript technically sound, and do the data support the conclusions?

Reviewer #5: Yes

Reviewer #6: (No Response)

3. Has the statistical analysis been performed appropriately and rigorously? 

Reviewer #5: Yes

Reviewer #6: (No Response)

4. Have the authors made all data underlying the findings in their manuscript fully available?

Reviewer #5: Yes

Reviewer #6: (No Response)

5. Is the manuscript presented in an intelligible fashion and written in standard English?

Reviewer #5: Yes

Reviewer #6: (No Response)

6. Review Comments to the Author

Reviewer #5: I would like to thank the authors for their thoughtful and thorough revisions. The manuscript has been significantly improved and all major concerns have been adequately addressed. I have no further suggestions. I recommend the manuscript for acceptance in its current form.

Reviewer #6: (No Response)

7. PLOS authors have the option to publish the peer review history of their article (what does this mean? ). If published, this will include your full peer review and any attached files.

**Do you want your identity to be public for this peer review?** For information about this choice, including consent withdrawal, please see our Privacy Policy .

Reviewer #5: No

Reviewer #6: No

---

## [Editor Report · Acceptance letter]

PONE-D-24-34699R1

PLOS ONE

Dear Dr. Al-Mamun,

I'm pleased to inform you that your manuscript has been deemed suitable for publication in PLOS ONE. Congratulations! Your manuscript is now being handed over to our production team.

Kind regards,

on behalf of

Dr. Wisit Kaewput

Academic Editor

PLOS ONE